# Genetic Features of Triticale–Wheat Hybrids with Vaviloid-Type Spike Branching

**DOI:** 10.3390/plants11010058

**Published:** 2021-12-25

**Authors:** Irina G. Adonina, Andrey B. Shcherban, Maremyana V. Zorina, Sabina P. Mehdiyeva, Ekaterina M. Timonova, Elena A. Salina

**Affiliations:** 1Institute of Cytology and Genetics SB RAS, Lavrentiev Av., 10, 630090 Novosibirsk, Russia; atos@bionet.nsc.ru (A.B.S.); mariamnazorina@gmail.ru (M.V.Z.); eegorova@bionet.nsc.ru (E.M.T.); salina@bionet.nsc.ru (E.A.S.); 2Kurchatov Genomic Center, Institute of Cytology and Genetics SB RAS, Lavrentiev Av., 10, 630090 Novosibirsk, Russia; 3Department of Natural Sciences, Novosibirsk State University, Pirogova Str., 1, 630090 Novosibirsk, Russia; 4Genetic Resources Institute of ANAS, Azadlig Ave., 155, Baku AZ1106, Azerbaijan; mora-kasper@rambler.ru

**Keywords:** triticale–wheat hybrid lines, vaviloid type of spike branching, sham ramification, karyotype, gene *Q*, APETALA2 (AP2)-like transcription factors

## Abstract

Vaviloid spike branching, also called sham ramification, is a typical trait of *Triticum vavilovii* Jakubz. and is characterized by a lengthening of the spikelet axis. In this article, we present the results of a study of three triticale–wheat hybrid lines with differences in terms of the manifestation of the vaviloid spike branching. Lines were obtained by crossing triticale with hexaploid wheat, *T. aestivum* var. *velutinum*. The parental triticale is a hybrid of synthetic wheat (*T. durum* × *Ae. tauschii* var. *meyrei*) with rye, *S. cereale* ssp. *segetale*. Line 857 has a karyotype corresponding to hexaploid wheat and has a spike morphology closest to normal, whereas Lines 808/1 and 844/4 are characterized by the greatest manifestation of vaviloid spike branching. In Lines 808/1 and 844/4, we found the substitution 2RL(2DL). The karyotypes of the latter lines differ in that a pair of telocentric chromosomes 2DS is detected in Line 808/1, and these telocentrics are fused into one unpaired chromosome in Line 844/4. Using molecular genetic analysis, we found a deletion of the wheat domestication gene *Q* located on 5AL in the three studied hybrid lines. The deletion is local since an analysis of the adjacent gene *B1* showed the presence of this gene. We assume that the manifestation of vaviloid spike branching in two lines (808/1 and 844/4) is associated with a disturbance in the joint action of genes *Q* and *AP2L2-2D*, which is another important gene that determines spike morphology and is located on 2DL.

## 1. Introduction

The *Triticum* L. genus includes the most important agricultural crops, namely, common wheat (*T. aestivum* L., BBAADD, 2n = 6x = 42) and durum wheat (*T. durum* Desf., BBAA, 2n = 4x = 28). Since wheat is a grain crop, the morphology of its inflorescence is of particular importance. A standard wheat inflorescence is a complex spike that consists of an axis and spikelets located on its ledges, with one spikelet per ledge. There are 3–5 florets in a spikelet, and the top 1–2 florets are usually sterile. Forms with nonstandard spike morphology are interesting in connection with the potential for increasing yields. Therefore, it is especially important to understand the genetic determination of inflorescence development. *WHEAT FRIZZY PANICLE (WFZP)* gene in wheat is one of the key regulators of inflorescence development [1,2]. It encodes transcription factors of the APETALA2/Ethylene Response Factor (AP2/ERF) family [3]. Coding mutations of homoeologous genes *WFZP-A*, especially *WFZP-D*, localized on the short arms of chromosomes 2A and 2D of common wheat, cause the supernumerary spikelet (SS) phenotype when the spike axis can branch and additional spikelets form on it or several spikelets form at the ledge of the spike [1,4]. Poursarebani et al. [2] has suggested that a single amino acid substitution in the AP2/ERF domain of *TtBH-A1*, which represents a mutant allele at the *WFZP-A* locus, causes spike branching in tetraploid wheat *(T. turgidum* convar. *compositum* (L.f.) Filat.). Dobrovolskaya et al. [1] reported that *WFZP*-*B* homoeologous gene (localized on 2BS) is practically inactive due to the miniature inverted-repeat transposable elements (MITE) insertion in the promoter region. However, Wolde et al. [5] showed that despite the inserted MITE close to *WFZP-B/**TtBH*-*B1*, the homoeo-gene is still expressed in developing spike tissues in tetraploid wheat and can function as a modifier for spike branching in tetraploid wheat.

Vaviloid-type spike branching or sham ramification [6] is a trait of the species *Triticum vavilovii* (Thum.) Jakubz. (2n = 6x = 42, BBAADD). In this case, additional spikelets on the axis node are not developed but the spikelet axis is lengthened, on which many florets are formed [7]. Sham ramification was also found in tetraploid wheat *T. jakubzineri* Udacz. et Schachm (2n = 4x = 28, BBAA) [8]. Aliyeva and Aminov [9,10] found that forms with a vaviloid type of spike branching can occur as a result of distant hybridization, and they showed that the spike branching of the vaviloid type (sham ramification) in hybrid wheat Line 166-Schakheli (2n = 4x = 28, BBAA) was controlled by one recessive gene [9]. Amagai [8] localized the *sham ramification 1* (*shr1*) gene of *T. jakubzineri* on the long arm of chromosome 5A in a similar position to gene *Q*, which is considered one of the main genes of wheat domestication. The *Q* allele is present in most cultivated wheat, and it controls the subcompact spike phenotype, rachis fragility, glume toughness, plant height, and flowering time. The *q* allele is associated with a speltoid spike (a spear-shaped spike with an elongated rachis) and nonfree-threshing grains [11]. The *Q* gene refers to APETALA2 (AP2)-like transcription factors [12]. In turn, a family of *APETALA2* (*AP2*) transcription factor genes is the regulatory target of microRNA172 (miR172). It has been shown that decreases in the expression of the *Q* gene or overexpressing miR172 lead to the formation of sham ramification [13,14]. Recently, Debernardi et al. [15] showed that the *Q* gene (*AP2L5*) and its related paralogue *AP2L2* play critical and redundant roles in the specification of axillary floral meristems.

In our work, we studied triticale–wheat hybrid lines with differences in terms of the manifestation of the vaviloid type of spike branching. These lines were obtained from crossing a wheat–rye amphiploid (triticale) (2n = 6x = 42, BBAARR), which was used as the maternal form, with the common wheat *T. aestivum* var. *velutinum* (2n = 6x = 42, BBAADD). Triticale was obtained from hybridization of synthetic wheat (*T. durum* × *Ae. tauschii* var. *meyeri*, 2n = 6x = 42, BBAADD) with weed rye *Secale cereale* ssp. *segetale* (2n = 2x = 14, RR). The forms with the vaviloid type of spike branching began to appear from generation F4. However, the genomic structure of these lines has not been explored prior to our study. We combined phenotypic, cytological and molecular genetic analyses to clarify the genetic nature of vaviloid-type spike branching in these lines.

## 2. Results

### 2.1. Genome Structure of Triticale–Wheat Hybrids

Earlier, we found that the triticale–wheat hybrid lines with the vaviloid type of spike branching from the collection of the Genetic Resources Institute of ANAS (Azerbaijan) are heterogeneous in karyotypic characteristics (the number of chromosomes, the presence of telocentrics) [16]. Therefore, we chose contrasting lines in terms of the manifestation of the spike branching trait and described the individual plants of each line.

Fluorescence in situ hybridization (FISH) with probes pSc119.2 [17] and pAs1 [18] allowed us to identify chromosomes of the B and D subgenomes of wheat (Figure 1d, e). The karyotype of Line 857 corresponded to the karyotype of hexaploid wheat (2n = 6x = 42, BBAADD) (Figure 1e). Lines 808/1 (Figure 1a,b) and 844/4 (Figure 1c, d) were characterized by the absence of 2D chromosomes in all analyzed plants. Genome in situ hybridization (GISH) with *S. cereale* DNA confirmed the absence of rye translocations in Line 857 and revealed the substitution of chromosome 2D by rye chromosome in Lines 808/1 (Figure 1a) and 844/4 (Figure 1c). We assumed that this chromosome was 2R considering that substitutions usually occur between homoeologous chromosomes. However, the rye chromosomes looked shorter than the normal chromosomes. A FISH analysis on metaphase chromosomes of Lines 808/1 and 844/4 with centromeric probes pAwRc [19] and pAet6-09 [20] (Figure 1b) demonstrated that rye chromosomes are telocentrics; thus, they were likely 2RL chromosomes (Figure 1h). In plants of Line 808/1, we found an additional pair of telocentric chromosomes of wheat (Figure 1a,b), and according to the length of the arm and the location of the probe pSc119.2 (Figure 1e,f), we assumed that these chromosomes were 2DS. The analyzed plants of Line 844/4 had 43 chromosomes (Figure 1d). An unpaired chromosome is formed by the centric fusion of two wheat telocentrics (Figure 1d,g).

To test the assumption about the presence of chromosomes 2RL and 2DS in Lines 808/1 and 844/4, we carried out molecular genetic analysis with chromosome-specific markers F3h-R1 [21] and Ppd-D1 [22] (Table 1).

When the 2RL-specific marker was used (Figure 2a), a marker fragment of DNA was present only in rye, triticale, and lines with rye telocentric chromosomes (808/1 and 844/4); in wheat and Line 857 with a karyotype of hexaploid wheat, the marker fragment was not amplified. When we used a 2DS-specific marker (Figure 2b), a marker fragment was present in wheat Chinese Spring and in hybrid lines: Line 857 with a karyotype of hexaploid wheat, Line 808/1 with a pair of wheat telocentric chromosomes, and Line 844/4 with an unpaired chromosome formed by the centric fusion of two wheat telocentrics. The marker fragment was not amplified in rye and triticale.

It should be noted, that simultaneously with us, Professor A. Lukaszewski (University of California, United States, personal communication) showed the presence of 2RL chromosomes in other two lines of the same parental crossing combination by C-banding.

### 2.2. Analysis of Spike Morphology in Different Karyotypes

We evaluated the phenotype and karyotype of fifteen plants of Line 808/1, ten plants of Line 844/4, thirteen plants of Line 857 and five plants of the parental triticale. The presence of telocentric chromosomes and an unpaired chromosome can cause cytological instability. However, all plants of the same line had the same karyotype. The main distinctive features of a vaviloid spike are lengthening of the spikelet axis and a greater number of florets in a spikelet. We evaluated these parameters primarily for the studied lines. Summarized comparison data of spike morphology and plant karyotype are given in Table 2.

The spike morphology in Line 857 with a karyotype corresponding to hexaploid wheat is the closest to normal. Lines 808/1 and 844/4 are characterized by 2RL(2D) substitution. The karyotypes of these lines differ in that a pair of telocentric chromosomes 2DS is detected in Line 808/1, and these telocentrics are fused into one unpaired chromosome in Line 844/4. Vaviloid-type spike branching is more pronounced in plants of Line 808/1 (Figure 3a, Figure 4) than in plants of Line 844/4 (Figure 3b, Figure 4), and these differences are significant. A total of 71–100% of spikelets were elongated in the primary spike of plants in Line 801/1, whereas 22–83% were elongated in Line 844/4.

An evaluation of the grain number from the primary spike (Figure 5) showed that despite the large number of florets, plants with a vaviloid type of spike branching had low productivity. Most of the florets were sterile, and grain formation was mainly at the base of the elongated spikelets.

### 2.3. Gene Q Analysis

The important role of the *Q* gene located on chromosome 5A of *T. aestivum* in determining spike morphology was previously established [12,13,23]. To identify possible allelic variants of this gene, we used four combinations of primers for the promoter and coding (within 8–10 exons) regions of the gene. In the latter case (QTaF/QTaR and QTsF/QTsR), a region between two single nucleotide polymorphisms (SNPs) was used to allow us to distinguish the forms with the normal spike, which is a characteristic of cultivated hexaploid wheat, from the forms with the speltoid spike, which is a characteristic of *T. spelta* L. and other *Triticum* species, predominantly wild. In the forward primers (QTaF and QTsF), the SNP leads to the replacement of the amino acid isoleucine by valine (in speltoid forms) (Table 1). In the reverse primers (QTaR and QTsR), the SNP affects the miRNA172 binding site, which could play a possible role in the regulation of *Q* gene expression by miRNA [12].

The PCR results with primers specific to the *Q* gene are shown in Figure 6. All hybrid lines (808/1, 844/4, 857) had no PCR products with any of the primer combinations. The sample *T. aestivum* Chinese Spring and the triticale samples contain dominant and recessive alleles of the *Q* gene that correspond to their normal and speltoid forms of spike, respectively (Figure 6b,c). The recessive allele is also present in rye, which indicates its ancient origin. In the case of common primers for the promoter region and exon 10, all of the above samples showed the same PCR products for each combination (Figure 6a,d). The absence of any amplification products in the hybrid lines implies the deletion of the *Q* gene.

To check whether this deletion is local or involves the whole terminal region of the 5AL chromosome, including the *Q* gene, we chose primer pairs specific to the *B1* gene, which determine awned/awnless forms of wheat (Table 1). This gene has recently been identified and is located more distantly from the *Q* gene on 5AL (at a distance of approximately 43 Mb) [24]. The result of amplification with two primer pairs showed the presence of PCR products in all hybrids: in the case of awned Lines 808/1 and 844/4 with primers b1for/Znfrev and awnless Line 857 with primer B1for/Znfrev (Figure 7). This indicates that if the deletion is present on 5AL in the studied hybrids, it is interstitial and involves the region of the *Q* gene but not the distal part of the chromosome where the *B1* gene is located.

## 3. Discussion

The vaviloid branching of a spike has been a subject of interest for a long time because similar traits allow the formation of more florets in a spikelet and, consequently, are potentially important for increasing productivity. The development of this trait in hexaploid wheat is poorly studied due to the complex structure of their genomes, which consist of three subgenomes. Singh et al. [25] suggested that the vaviloid character is determined by two genes and that one of these genes is closely linked or identical to the gene *Q*. Prabhakara-Rao and Swaminathan [26] obtained a vaviloid mutant in *T. aestivum* and hypothesized that the gene *Q* suppressed the expression of the vaviloid character in *T. aestivum*. Amagai et al. [8] localized the *sham ramification 1* (*shr1*) gene of *T. jakubzineri* in close vicinity of the *extra glume* (*exg*) gene and microsatellite marker *Xbarc319* on the long arm of chromosome 5A. Earlier, Kosuge et al. [27] showed that the marker *Xbarc319* flanks the compact spike gene *C17648.* Finally, Greenwood et al. [14] revealed that the phenotype of lines with compact spikes is associated with certain SNPs within the miRNA172 binding site of gene *Q*. Thus, it can be assumed that the compact spike gene *C17648, exg* gene, and *shr1* gene are in fact allelic variants of the same *Q* gene.

Currently, it is believed that the key mechanism determining the development and architecture of inflorescences in cereals includes the interaction of miRNA172 with a target, the gene encoding an AP2-like transcription factor, in particular, the *Q* gene [12,28]. More recently, it has been shown that decreases in the expression of the *Q* gene or overexpressing miR172 lead to the formation of vaviloid branching of a spike [13,14]. The same authors describe loss-of-function alleles of the *Q* gene that were sufficient to induce elongated spikelets with additional florets on the spikelet axis. We failed to identify gene *Q* in our hybrid lines using PCR with different primer combinations (Figure 6), but we obtained amplification in the case of the *B1* gene (Figure 7) located more distantly, at the terminal part of chromosome 5AL. These results imply a local deletion of the *Q* gene in the studied hybrids. Interestingly, Vavilova et al. [29] did not obtain positive results during PCR amplifications of the *Q* gene in the accessions of *T. vavilovii*, which is the species for which this type of spike branching was named. This result may also indicate a gene *Q* deletion in *T. vavilovii*.

The hybrid lines studied in our work have a complex origin. One of the parental forms was triticale, a hybrid of synthetic wheat with rye (see the Section Plant Material). This parental form has an intact recessive *q* allele based on PCR analysis (Figure 6). Unfortunately, we cannot check whether the second parent *T. aestivum* var. *velutinum* has an intact *Q* gene because this variety is currently not in our collection. It seems obvious to us that the observed *Q* gene deletion occurred during hybridization in the process of chromosome recombination since both parents had a normal phenotype.

Alieva and Aminov [10] suggested that the presence of the D subgenome suppresses the manifestation of the vaviloid branching of a spike because this trait appeared only in hybrid plants that do not carry the D subgenome, although the specific chromosomes of subgenome D involved in this trait have not been identified. Dobrovolskaya et al. [1] previously showed particular importance of the *WFZP-D* gene localized on the 2DS chromosome in the regulation of wheat spike development and the emergence of forms with a branched spike. In the present work, we have shown that the lines with the most pronounced vaviloid branching of a spike (808/1 and 844/4) carried telocentric chromosome 2DS. In Line 844/4, these telocentrics formed an unpaired chromosome by centric fusion. Therefore, if the *WFZP-D* gene has not undergone changes during hybridization, then it seems unlikely that it plays a significant role in the development of the studied trait, since in different lines with manifestations of vaviloid branching, the 2DS chromosome inherited from the parent without branching is present.

Changes in spike morphology were coupled with the substitution 2RL(2DL). Only plants of Line 857, which had no last substitution and displayed the karyotype of hexaploid wheat with intact chromosome 2D, had a spike morphology close to normal. Therefore, it can be assumed that a long arm of chromosome 2D that is substituted in Lines 808/1 and 844/4 by 2RL suppresses the manifestation of vaviloid branching.

Ning et al. [30] described members of the AP2 gene family designated *TaAP2*, one of which, *TaAP2-D*, has been mapped on the subtelomeric region of chromosome 2DL. Recently, Debernardi et al. [15] showed that *AP2L2* together with *Q* (*AP2L5*) play an overlapping role in the regulation of floral axillary meristems. Loss-of-function mutants in homoeologs of *AP2L2* developed normal spikelets, but double mutants in both *AP2L2* and *Q* generated spikelets with multiple empty bracts before transitioning to florets. Morphologically, the spikes in our hybrids resemble the above mutants because most of the florets were sterile despite the large number of florets. This phenotype was very similar to the strongest wheat miR172 lines, which suggests that *AP2L2* and *AP2L5* account for most of the effect of the miR172-targeted genes involved in this function [15]. Debernardi and coauthors [15] studied the tetraploid wheat variety Kronos and investigated homoeologs *AP2L2-2A* and *AP2L2-2B*. The expression level of *AP2L2-2D* in hexaploid wheat is higher than that of *AP2L2-2A* and *AP2L2-2B* (15, http://www.wheat-expression.com, accessed on 18 October 2021); therefore, we can assume that spike morphology was close to normal in Line 857 with *Q* gene deletion due to the presence of *AP2L2-2D* on the long arm of intact chromosome 2D. Based on our findings and previous results, it can be suggested that the balance in the expression of AP2-like genes located on chromosomes 5AL (*Q*), 2AL (*AP2L2-2A*), 2BL (*AP2L2-2B*) and 2DL (*AP2L2*) is crucial for the correct development of spikelets and florets. This balance may be altered during the process of hybridization in cases when the given chromosomes undergo structural reorganization. Thus, the manipulation of this regulatory module provides an opportunity to modify spikelet architecture and perhaps improves grain yield.

## 4. Materials and Methods

### 4.1. Plant Material

We used three triticale–wheat hybrid lines with differences in terms of the manifestation of the vaviloid type of spike branching as a material. These lines (808/1; 844/4 and 857) were kindly provided by the Genetic Resources Institute of ANAS (Azerbaijan) and were obtained by crossing triticale with hexaploid wheat *T. aestivum* var. *velutinum*. Triticale is a hybrid of synthetic wheat BBAADD (*T. durum* × *Ae. tauschii* var. *meyrei)* with rye, *Secale cereale* ssp. *segetale* (2n = 2x = 14, RR). To estimate spike morphology, we cultivated the plants under greenhouse conditions (“Laboratory of Artificial Plant Cultivation” Centre) at the Institute of Cytology and Genetics, Novosibirsk, Russia, after vernalization for 60 days at a temperature of +4 °C.

### 4.2. Phenotypic Analysis

The phenotypes of the primary spikes of individual plants were described after full maturation. The phenotypic analysis included measurements of the following parameters: number of spikelets, number of elongated spikelets, spikelet length, number of florets per spikelet and number of grains per spike. We analyzed all spikelets in the spikes except for a pair of lower usually underdeveloped spikelets.

The data were statistically evaluated by one-way ANOVA and Student’s *t*-test.

### 4.3. Cytogenetic Analysis

We conducted FISH with probes based on the cloned DNA repeats according to a method published earlier [31]. Probes pSc119.2 [17] and pAs1 [18] were used to identify individual chromosomes; probes pAwRc [19] and pAet6-09 [20] were employed to distinguish centromeric regions. pAwRc and pAet6-09 clones were kindly provided by Professor A. Lukaszewski (University of California, United States). GISH with rye DNA (*S. cereale*) was carried out according to a published protocol [32]. The probes were labelled with biotin (biotin-16-dUTP, Roche) or digoxygenin (digoxigenin-11-dUTP, Roche) using Nick Translation Mix (Roche). Detection of the biotinylated probes was carried out using fluorescein avidin (fluorescein avidin D, Vector Laboratories, Burlingame, CA, USA). The hybridization signal was amplified using fluorescein anti-avidin (fluorescein anti-avidin D, Vector Laboratories, Burlingame, CA, USA). Digoxigenin-labelled probes were detected using Antidigoxigenin-rhodamine Fab fragments (Antidigoxigenin_rhodamine Fab fragments, Roche). The samples were embedded in fluorescence fading-inhibiting medium (Vectashield mounting medium, Vector Laboratories, Burlingame, CA, USA) containing 0.5 mg/mL DAPI (4′,6-diamidino-2-phenylindol, Sigma–Aldrich) for chromosome staining and analyzed using an Axio Imager M1 microscope (Zeiss, Oberkochen, Germany) equipped with a ProgRes MF CCD camera (Meta Sistems, Altlussheim, Germany) and Isis (Meta Sistems, Altlussheim, Germany) software.

### 4.4. DNA Isolation and Molecular Genetic Analysis

Total DNA was isolated from 5- to 7-day-old seedlings according to Plaschke et al. [33]. To identify the presence of the 2RL chromosome arm in the hybrids, we used a sequence-tagged site (STS) marker for the *F3h-R1* gene [21] (Table 1). To identify the presence of the 2DS chromosome arm, we used a marker for the *Ppd-D1* gene [22] (Table 1).

### 4.5. Gene Q Analysis

Four pairs of primers were used to amplify different regions of the *Q* gene (Table 1). We also used primers for gene *B1*, localized closer to telomere at a distance of approximately 43 Mb from the *Q* gene [24] (Table 1).

The reaction mixture contained 50–100 ng of DNA template, 1 × PCR buffer, 2 mM MgCl_2_, 0.2 mM of each dNTP, 0.5 mM of each primer, and 1 U of HS-Taq DNA polymerase (BiolabMix, Novosibirsk, Russia). The touch-down PCR program had an initial strand separation step at 95 °C for 3 min followed by 12 cycles of denaturation at 95 °C for 10 s, annealing at 65 °C for 20 s and elongation at 72 °C for 0.2–1 min (according to the expected size of the PCR product), followed by 25 similar cycles using an annealing temperature of 55 °C.

PCR products were separated in a 1% agarose gel containing ethidium bromide. After electrophoresis, DNA fragments were visualized and photographed in UV light using Gel Doc™ XR+ (Bio–Rad Laboratories, Inc., Hercules, CA, USA).

## 5. Conclusions

A cytological analysis of the hybrid lines obtained by crossing triticale with hexaploid wheat showed that lines with vaviloid spike branching are characterized by chromosomal substitution 2RL(2DL). A molecular genetic analysis of the *Q* gene (5AL), one of the main genes that determines spike morphology, revealed its complete deletion in the studied lines. We suggested that this deletion occurred during hybridization in the process of chromosome recombination. Previous studies have shown that other AP2-like genes, namely, *AP2L2*, are involved in the regulation of the floral axillary meristems, and one of the homoeologs (*AP2L2-2D*) was localized on the subtelomeric region of chromosome 2DL. Since our lines with vaviloid spike branching have the 2RL(2DL) substitution, and Line 857 presents a spike morphology close to normal without the last substitution and displays the karyotype of hexaploid wheat with an intact chromosome 2D, we assume that genes *Q* and *AP2L2-2D* jointly act in the suppression of vaviloid spike branching.

## Figures and Tables

**Figure 1 plants-11-00058-f001:**
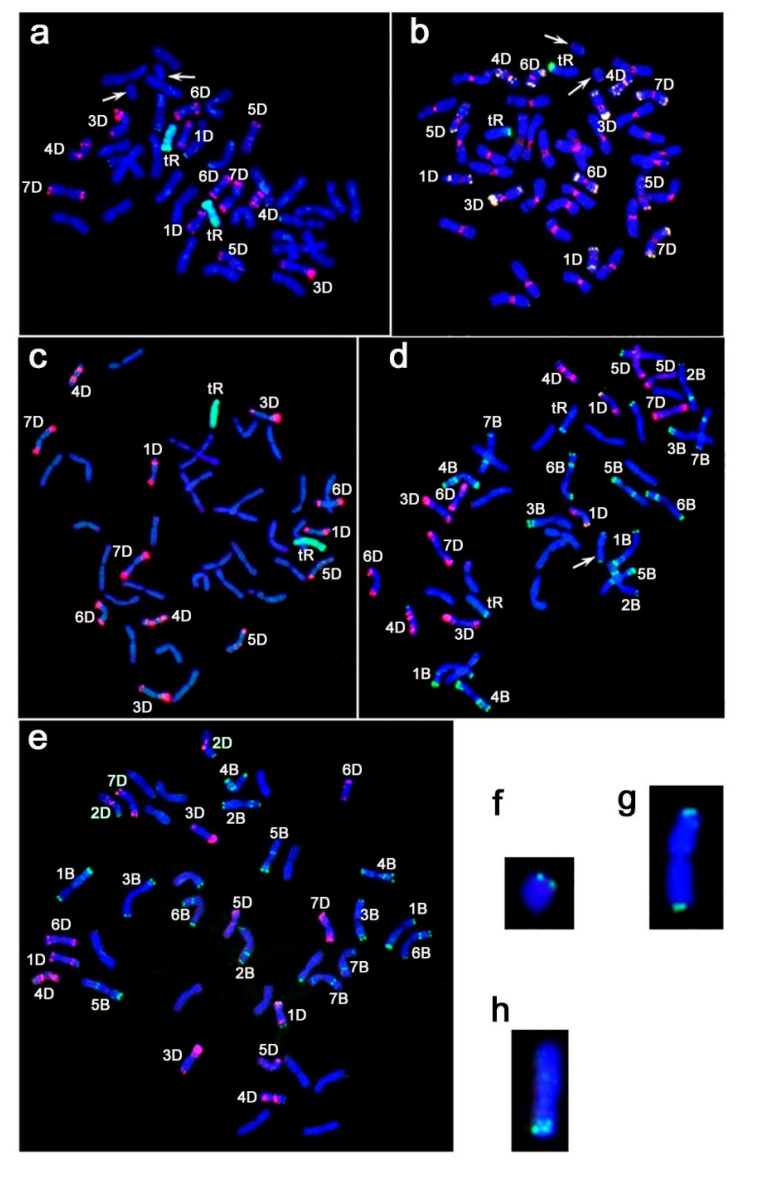
Fluorescence in situ hybridization (FISH) on mitotic metaphase chromosomes of the Line 808/1 plants (**a**,**b**); 844/4 (**c**,**d**) and 857 (**e**) with combinations of probes: (**a**) and (**c**)—pAs1 (red) and rye DNA (green); **b**—pAet6-09 (red), pAwRc (green) and pAs1 (yellow); (**d**) and (**e**)—pSc119.2 (green) and pAs1 (red) chromosomes are blue as a result of counterstaining with DAPI. Arrows indicate telocentric chromosomes of wheat (**a**,**b**) and an unpaired chromosome formed by the centric fusion of two wheat telocentrics (**d**). Separately presented: telocentric chromosomes of wheat (**f**), unpaired chromosome formed by the centric fusion of two wheat telocentrics (**g**) and telocentric chromosomes of rye (**h**). Green signals on these chromosomes indicate probe pSc119.2.

**Figure 2 plants-11-00058-f002:**
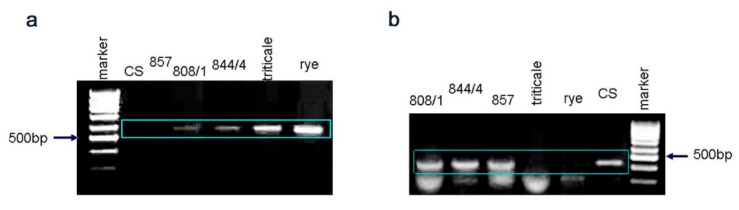
PCR analysis of the hybrid lines parental triticale, rye and common wheat Chinese Spring with the markers F3h-R1 (**a**) and Ppd-D1 (**b**).

**Figure 3 plants-11-00058-f003:**
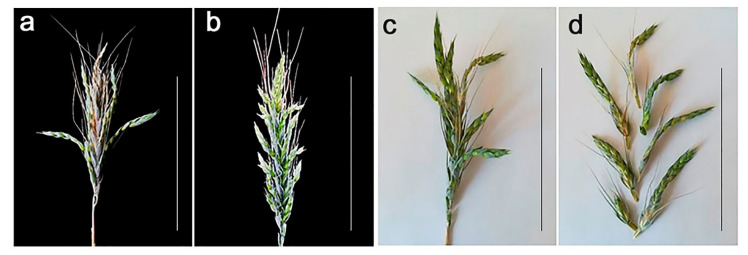
Vaviloid type of spike branching. Spike of Line 808/1 plant (**a**); spike of Line 844/4 plant (**b**); intact spike of Line 808/1 plants (**c**); and the same spike divided into spikelets (**d**). Scale bar: 10 cm.

**Figure 4 plants-11-00058-f004:**
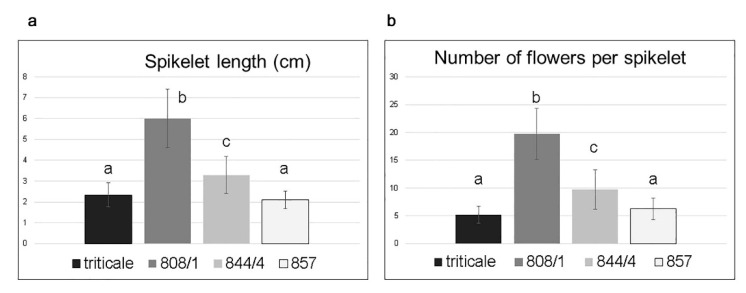
Main parameters of spikes in the studied lines: spikelet length (**a**), and number of flowers per spikelet (**b**). Different letters indicate statistically significant differences (*p* < 0.01, Student’s *t*-test).

**Figure 5 plants-11-00058-f005:**
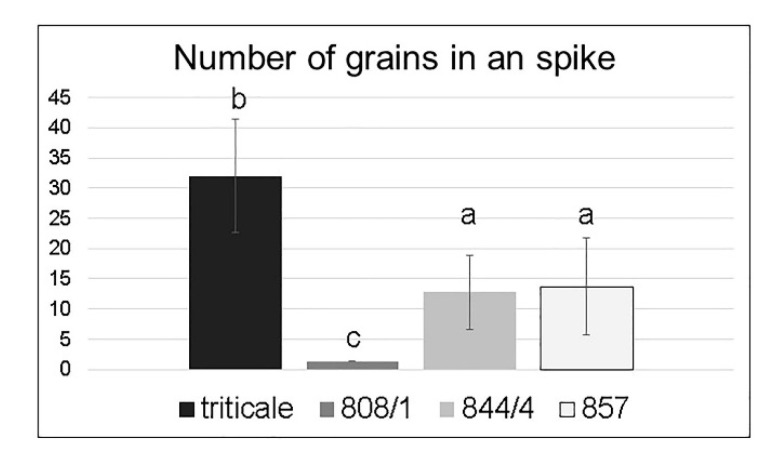
Number of grains per primary spike in the studied lines. Different letters indicate statistically significant differences (*p* < 0.01, Student’s *t*-test).

**Figure 6 plants-11-00058-f006:**
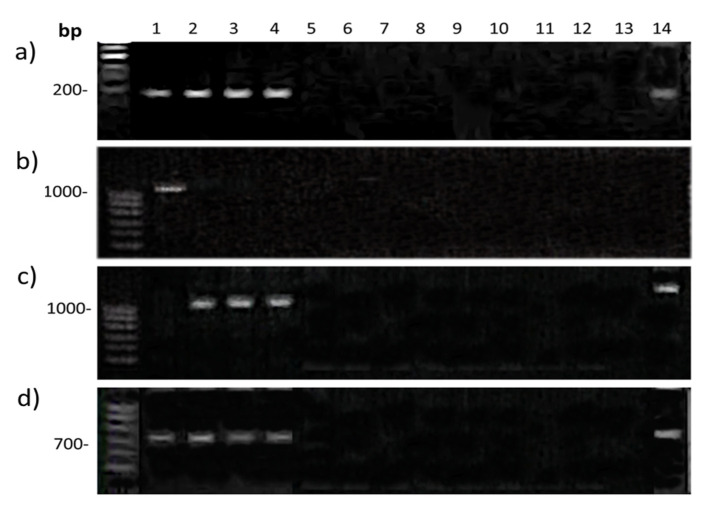
PCR amplification with primers: Qex10f/Qex10r (**a**); QTaF/QTaR (**b**); QTsF/QTsR (**c**); and Qpromf/Qpromr (**d**). Samples: 1—*T. aestivum* Chinese Spring; 2–4—triticale; 5–7—hybrid Line 808/1; 8–10—hybrid Line 844/4; 11–13—hybrid Line 857; and 14—*S. cereale*.

**Figure 7 plants-11-00058-f007:**
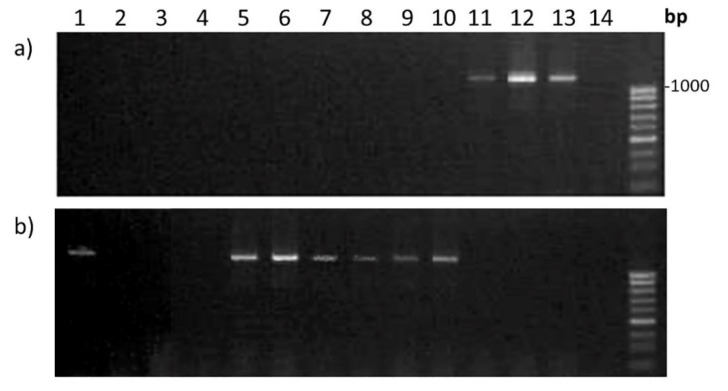
PCR amplification with primers B1for/Znfrev (**a**) and b1for/Znfrev (**b**). The first pair marks the awnless forms with the dominant *B1* gene, the second pair marks the awned forms with the recessive *b1* gene. Samples: 1—*T. aestivum* Chinese Spring; 2–4—triticale; 5–7—hybrid Line 808/1; 8–10—hybrid Line 844/4; 11–13—hybrid Line 857; and 14—*S. cereale*.

**Table 1 plants-11-00058-t001:** Primers used in the work.

Primer Name	Sequence 5′–3′	Gene	Chromosomal Localization	Expected PCR Product Size (bp)
F3h–F	CTGGGCGTGCTATCGGAGGT	*Rye flavanone 3-hydroxylase* *(F3h-R1)*	2RL	580
F3h–R	TTGGCGAGGTCGAGGTCGCGCTT			
Ppd_F	ACGCCTCCCACTACACTG	*Photoperiod response* (*Ppd-D1)*	2DS	414, 288
Ppd_R1	GTTGGTTCAAACAGAGAGC			
Ppd_R2	CACTG-GTGGT-AGCTG-AGATT			
Qex10F	AGGCAAGGCCCCCTGAGCA	*Q* * exon 10	5AL	175
Qex10R	CGGTGGTGGTCCGGGTACGG			
QTaF	CCCTGAATCGTCAACCACAATGA	exon 8–10 (*Q* allele)	5AL	1059
QTaR	CGCCGGCGGCGGCGGTAGAA			
QTsF	CCCTGAATCGTCAACCACAATGG	exon 8–10 (*q* allele)	5AL	1059
QTsR	CGCCGGCGGCGGCGGTAGAG			
QpromF	TGATGTACGCTCCGTGTGA	promoter region	5AL	709
QpromR	TGGAGGACGACGAGGAGAG			
B1for	ATAAACTCCCACATAATTACTTCG	*B1* * promoter region	5AL	1177
b1for	AAACTCCCACATAATTACTCCC			
Znfrev	CTCTTCCATCTCCATGCCCA			

* All primers were designed by the authors.

**Table 2 plants-11-00058-t002:** Comparison data of triticale–wheat hybrid line karyotypes and spike morphology.

Lines	Karyotype	Phenotype	
		Spikelet Length, cm	Number of Florets per Spikelet
		Mean (Range) of	Mean (Range) of
808/1	14A + 14B + 12D+2(2DS) + 2(2RL) = 44	6.0 (2.6–8.3) b	19.7 (7–27) b
844/4	14A + 14B + 12D + 2(2DS) (centric fusion) + 2(2RL) = 43	3.3 (1.0–5.4) c	9.7 (3–19) c
857	14A + 14B + 14D = 42	2.1 (1.0–3.2) a	6.2 (2–11) a
triticale	14A + 14B + 14R = 42	2.3 (0.9–3.8) a	5.2 (2–8) a
*T. aestivum* [7]	14A + 14B + 14D = 42	0.9–1.7	2–5

Note: for each line, the number of chromosomes is given in the column “Karyotype”. A, B and D indicate chromosomes of the three wheat subgenomes; R indicates the rye chromosome; RL indicates the telocentric chromosome of rye, representing its long arm; and DS indicates the telocentric chromosome of wheat, which is its short arm. Triticale and wheat are included for comparison. Different letters indicate statistically significant differences (*p* < 0.01, Student’s *t*-test).

## Data Availability

Not applicable.

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
