# Peer review of "Genetic Features of Triticale–Wheat Hybrids with Vaviloid-Type Spike Branching"

_plants, 2021, doi:10.3390/plants11010058_

Round 1
Reviewer 1 Report
Spike type is an important character of wheat, and it is also an issue of great concern to wheat breeders. The authors combined phenotypic, cytological and molecular genetic analyses to clarify the genetic nature of vaviloid-type spike branching in three triticale-wheat hybrid lines.I can state that the article can be publish. This will arouse the interest and attention of Wheat Genetics and Breeding scholars. For one question, in Figure 7. the first pair marks the awnless forms with the dominant B1 gene, the second pair marks the awned forms with the recessive b1 gene. Chinese Spring and 857 line are awnless wheat materials. Why are different B1 genotypes detected in them?
Author Response
Thanks for your comments and question!
Answer to the question: For one question, in Figure 7. the first pair marks the awnless forms with the dominant B1 gene, the second pair marks the awned forms with the recessive b1 gene. Chinese Spring and 857 line are awnless wheat materials. Why are different B1 genotypes detected in them?
They are really have different B1 genotypes. The awness trait is determined not only by the B1 gene, but also by two other genes (Hd and B2), but to a lesser extent than the first. Chinese Spring genotype is HdHdb1b1B2B2 (see ref. 17: Yoshioka M. et al; Three dominant awnless genes in common wheat: Fine mapping, interaction and contribution to diversity in awn shape and length. Plos one 2017, 12(4), e0176148). Two any dominant genes determine awnless phenotype. This explains why CS with b1 allele (should be awned) in reality is awnless. However, among other cultivars of common wheat such a situation is rather rare because of rare occurrence of Hd and B2 alleles.
Reviewer 2 Report
Vaviloid spike branching is a typical trait of Triticum vavilovii Jakubz and is characterized by a lengthening of the spikelet axis. In this article, the authors present the results of a study of three triticale-wheat hybrid lines with differences in terms of the manifestation of the vaviloid spike branching. They also found a deletion of the wheat domestication gene Q located on 5AL in the three studied hybrid lines and assumed that the manifestation of vaviloid spike branching in two lines was associated with a disturbance in the joint action of genes Q and AP2L2-2D. The results provide important reference for further in-depth study of molecular mechanism of Vaviloid spike branching. However, insufficient molecular data was provided and several concerns must be addressed by the authors.
Major concerns:
- For Results 2.2, Analysis of spike morphology in different karyotypes. Vaviloid spike branching is a typical trait of Triticum vavilovii Jakubz. The authors should provide electron microscope pictures of the early development of these three hybrid lines’ inflorescences.
- For Results 2.3., Gene Q analysis. The authors should provide the expression patterns of Q gene and the other two homoeologous genes on the B and D subgenome here. Because Q is a dose-responsive gene, and the copy of the D genome may affect the function of 5AQ as well.
- For the data in Table 2 and Figure 5, statistical analysis should be performed.
Minor concerns:
- The decimal points between the numbers in Table 1 are written as commas;
- Bar is not marked in Figure 3.
Author Response
Thank you for the review, important comments and remarks!
Answers on remarks:
Major concerns:
- For Results 2.2, Analysis of spike morphology in different karyotypes. Vaviloid spike branching is a typical trait of Triticum vavilovii Jakubz. The authors should provide electron microscope pictures of the early development of these three hybrid lines’ inflorescences.
We do not currently have electron microscope pictures of the early development of hybrid lines’ inflorescences. We are going to do microscopic analysis at the same time as studying genes expression. The results of the microscopic analysis and analysis of gene expression will be presented in a separate publication.
- For Results 2.3., Gene Q analysis. The authors should provide the expression patterns of Q gene and the other two homoeologous genes on the B and D subgenome here. Because Q is a dose-responsive gene, and the copy of the D genome may affect the function of 5AQ as well.
For analysis of gene expression we are going to use RNAseq technology allowing to study not only the Q- gene, but also other putative genes that are associated with this trait. But such an analysis will require much more time than was allotted to correct the manuscript. And this will be the next stage, the results of which will be presented in a separate publication.
- For the data in Table 2 and Figure 5, statistical analysis should be performed.
We have added statistical analysis data in Table 2 and Figure 5.
Minor concerns:
- The decimal points between the numbers in Table 1 are written as commas.
Apparently, it meant Table 2. The commas are corrected to decimal points.
- Bar is not marked in Figure 3.
Bar is marked in Figure 3.
Reviewer 3 Report
The MS "Genetic features of triticale-wheat hybrids with vaviloid-type spike branching describes the probable genetic regulation of vaviloid spike branching. The authors conducted their cytological, phenotypic, and genetic analyses in amphiploid lines that were derived from a cross between triticale and hexaploid wheat. Overall the MS reads well except in certain areas (given in comments). Please find my comments on the MS.
Comments/suggestions
- The MS has language inconsistencies in certain parts of the text, especially the introduction and results sections. It would be helpful if the MS is proofread by a native English language speaker.
- The authors mentioned that WFZP-B on chromosome 2BS does not influence spike development (lines 40-41). However, Gizaw et al. 2021 (https://link.springer.com/article/10.1007/s00122-020-03743-5) showed that 2BS copy of WFZP functions as a modifier for spike branching in tetraploid wheat. Please check the literature.
- The authors must cite all relevant literature unless there is a word/page number limitation. The WFZP has been extensively studied by several researchers which should be cited, if not at least the major studies in this field.
- lines 58-59 - The APETAL 2 of Q is distinct from AP2-ERF encoded by WFZP. Please take care while reporting.
- It was difficult to understand why the 2DS primers did not amplify a fragment while the lines 808/1 and 844/4 still contained 2DS chromosome (Table 2). Please clarify.
- Spikelets from which part of the spike have been analyzed for the floret number and spikelet length differences between the parents and the vaviloid spike-types. Since there is a gradient in terms of spikelet axis elongation phenotype (more elongated in the spikelets from the mid-region of the spike compared to spikelets at the base and tip of the spike), this comparison should be ideally done in defined spike regions across parents and vaviloid spike-types.
- About the amplification of the B1 gene to confirm the Q gene deletion in 808/1 and 844/4, I could not understand, why the authors chose a gene that is located 48 Mb away from the Q gene. It would be better to confirm the deletion by checking the amplification of flanking genes of Q or essentially characterize the whole deletion by amplifying the genomic region around Q. This will enable to pinpoint whether the causal phenotype is due to the Q gene alone or some other genes around the Q locus that can influence the vaviloid phenotype.
- In monocots, flowers are termed florets. Please replace wherever necessary.
Author Response
Thank you for the valuable comments and important remarks!
Answers on comments/suggestions:
- The MS has language inconsistencies in certain parts of the text, especially the introduction and results sections. It would be helpful if the MS is proofread by a native English language speaker. A native English language speaker in American Journal Experts already proofread MS. I am attaching the certificate.
- The authors mentioned that WFZP-B on chromosome 2BS does not influence spike development (lines 40-41). However, Gizaw et al. 2021 (https://link.springer.com/article/10.1007/s00122-020-03743-5) showed that 2BS copy of WFZP functions as a modifier for spike branching in tetraploid wheat. Please check the literature. Corrections have been made to the text in accordance with your remark, and a link has been added (Lines: 50-53 and Lines: 378-380 - References).
- The authors must cite all relevant literature unless there is a word/page number limitation. The WFZP has been extensively studied by several researchers which should be cited, if not at least the major studies in this field. We have cited other researchers in the text of the manuscript (Lines: 39-53) and added links in the References (Lines: 369-380).
- Lines 58-59 - The APETAL 2 of Q is distinct from AP2-ERF encoded by WFZP. Please take care while reporting. We have made the appropriate corrections to the text (Lines: 40-41 and 67-68)
- It was difficult to understand why the 2DS primers did not amplify a fragment while the lines 808/1 and 844/4 still contained 2DS chromosome (Table 2). Please clarify. 2DS primers amplify a marker fragment in the lines 808/1 and 844/4. Lines 123-130: “When we used a 2DS-specific marker (Figure 2b), a marker fragment was present in wheat Chinese Spring and in hybrid lines: Line 857 with a karyotype of hexaploid wheat, Line 808/1 with a pair of wheat telocentric chromosomes and Line 844/4 with an unpaired chromosome formed by the centric fusion of two wheat telocentrics. The marker fragment was not amplified in rye and triticale.”
- Spikelets from which part of the spike have been analyzed for the floret number and spikelet length differences between the parents and the vaviloid spike-types. Since there is a gradient in terms of spikelet axis elongation phenotype (more elongated in the spikelets from the mid-region of the spike compared to spikelets at the base and tip of the spike), this comparison should be ideally done in defined spike regions across parents and vaviloid spike-types. We have been analyzed all spikelets except for a pair of lower usually underdeveloped spikelets. However, this is a very good remark, that there is a gradient in terms of spikelet axis elongation phenotype, and comparison should be done in defined spike regions. We will definitely take this into account in our future work. Clarification made to the text (Lines: 482-483).
- About the amplification of the B1 gene to confirm the Q gene deletion in 808/1 and 844/4, I could not understand, why the authors chose a gene that is located 48 Mb away from the Q gene. It would be better to confirm the deletion by checking the amplification of flanking genes of Q or essentially characterize the whole deletion by amplifying the genomic region around Q. This will enable to pinpoint whether the causal phenotype is due to the Q gene alone or some other genes around the Q locus that can influence the vaviloid phenotype. Of course, other genes located in the immediate vicinity of the Q gene may potentially influence the vaviloid phenotype. But to map the region of deletion it is necessary to design a number of primers combinations and to make a lot of PCRs. We made PCR with primers annealed about 3 kb upstairs the Q-gene and no PCR products were obtained (we can include this result in Figure 6 if necessary). Besides, suppose we found genes that were deleted along with Q. It will take a separate work to prove their participation in the studied trait determination. Since there are few genes with putative influence on this trait and there are definitely no homologous genes in the region between Q and B1.
- In monocots, flowers are termed florets. Please replace wherever necessary. “Flowers” are replaced by “florets” in the text and in the Figure4 (b).
